**Data Availability Statement:** The dataset is freely available for download at https://dhsprogram.com/data/available-datasets.cfm.

# Ending violence against women: Help-seeking behaviour of women exposed to intimate partner violence in sub-Saharan Africa

Richard Gyan Aboagye[1]*, Abdul-Aziz Seidu[2,3,4], Abdul Cadri[5,6], Tarif Salihu[7], Francis Arthur-Holmes[8], Sarah Tara Sam[8], Bright Opoku Ahinkorah[4,9,10]

1 Department of Family and Community Health, Fred N. Binka School of Public Health, University of Health and Allied Sciences, Hohoe, Ghana, 2 College of Public Health, Medical and Veterinary Services, James Cook University, Townsville, Australia, 3 Centre For Gender and Advocacy, Takoradi Technical University, Takoradi, Ghana, 4 REMS Consultancy Services, Sekondi-Takoradi, Western region, Ghana, 5 Department of Social and Behavioural Science, School of Public Health, University of Ghana, Legon, Accra, Ghana, 6 Department of Family Medicine, Faculty of Medicine, McGill University, Montreal, Quebec, Canada, 7 Department of Population and Health, University of Cape Coast, Cape Coast, Ghana, 8 Department of Sociology and Social Policy, Lingnan University, Tuen Mun, Hong Kong, China, 9 Academic Unit of Infant, Child, and Adolescent Psychiatry Services (AUCS), SWSLHD and Ingham Institute, 10 Discipline of Psychiatry and Mental Health, University of New South Wales, Sydney, Australia

* raboagye18@sph.uhas.edu.gh

## Abstract

### Introduction

Intimate partner violence is a serious public health problem that transcends cultural boundaries in sub-Saharan Africa. Studies have reported that violence characteristics and perception are strong predictors of help-seeking among women. We assessed the prevalence and factors associated with help-seeking among female survivors of intimate partner violence in sub-Saharan Africa.

### Methods

We pooled data from the most recent Demographic and Health Surveys (DHS) of eighteen sub-Saharan African countries. The data were extracted from the women's files in countries with datasets from 2014 to 2021. A weighted sample of 33,837 women in sexual relationships: married or cohabiting who had ever experienced intimate partner violence within the five years preceding the survey were included in the analysis. Percentages with 95% confidence interval (CI) were used to present the results of the prevalence of help-seeking for intimate partner violence. We used a multilevel binary logistic regression analysis to examine the factors associated with help-seeking among survivors of intimate partner violence. The results were presented using adjusted odds ratio (AOR) with their respective 95% CI. Statistical significance was set at p<0.05.

### Results

Out of the 33,837 women who had ever experienced intimate partner violence in sub-Saharan Africa, only 38.77% (95% CI = 38.26–39.28) of them sought help. Ethiopia had the

**Funding:** The author(s) received no specific funding for this work.

**Competing interests:** The authors have declared that no competing interests exist.

**Abbreviations:** AIC, Akaike Information Criterion; AOR, Adjusted Odds Ratio; CIs, Confidence Intervals; DHS, Demographic and Health Survey; EA, Enumeration Area; HIV, Human Immunodeficiency Virus; ICC, Intra-Class Correlation; IPV, Intimate Partner Violence; PSU, Primary Sampling Unit; SSA, sub-Saharan Africa; VIF, Variance Inflation Factor; WHO, World Health Organization.

lowest prevalence of women who sought help after experiencing intimate partner violence (19.75%; 95% CI = 17.58–21.92) and Tanzania had the highest prevalence (57.56%; 95% CI = 55.86–59.26). Marital status, educational level, current working status, parity, exposure to interparental violence, women's autonomy in household decision-making, mass media exposure, intimate partner violence justification, wealth index, and place of residence were associated with help-seeking behaviour of intimate partner violence survivors.

## Conclusion

The low prevalence of help seeking among women who have experienced intimate partner violence in sub-Saharan Africa calls for the intensification of formal and informal sources of assistance. Education can play a critical role in empowering girls, which may increase future help-seeking rates. Through media efforts aimed at parental awareness, the long-term benefits of females enrolling in school could be achieved. However, concentrating solely on individual measures to strengthen women's empowerment may not bring a significant rise in help-seeking as far as patriarchal attitudes that permit violence continue to exist. Consequently, it is critical to address intimate partner violence from the dimensions of both the individual and violence-related norms and attitudes. Based on the findings, there should be public awareness creation on the consequences of intimate partner violence. Respective governments must increase their coverage of formal support services to intimate partner violence survivors especially those in rural communities.

## Introduction

Intimate partner violence (IPV) is a serious public health problem that transcends cultural boundaries and takes different forms such as physical violence, sexual violence, stalking, and psychological aggression by a current or former intimate partner [1]. The World Health Organization (WHO) has estimated that, globally, about 30% of women have experienced IPV in their lifetime [2]. IPV has many negative implications on the health and wellbeing of women who experience it [3]. These implications include human immunodeficiency virus (HIV) and other sexually transmitted infections, induced abortion, low birth weight and preterm birth, harmful alcohol use, depression, suicide, non-fatal injuries, and fatal injuries [4]. Other effects of IPV have been reported to include adolescent pregnancy, unintended pregnancy, miscarriage, stillbirth, intrauterine haemorrhage, nutritional deficiency, abdominal pains and other gastrointestinal problems, neurological disorders, chronic pain, disability, anxiety, and post-traumatic stress disorder [5–7]. Though IPV is prevalent in most parts of the world, the sub-Sahara African regional level recorded the highest prevalence (approximately 33%) [2]. In recent times, there has been a growing interest in research on IPV among women in sub-Saharan Africa (SSA). However, the help-seeking behaviour of women who are exposed to IPV is understudied at the sub-Sahara African regional level.

Studies have reported that female survivors of IPV who seek support have a lower risk of further violence and lower odds of experiencing depression and have higher self-esteem [8, 9]. Despite the high prevalence of IPV among women in SSA [2], help-seeking for IPV among women in SSA is low. For instance, a study by Mahenge and Stöckl [10] reported that among the 41.6% of respondents who were survivors of IPV in Tanzania, only half of them sought help. Also, Tenkorang, Sedziafa, and Owusu [11] reported in their study that 65% of women

who experienced IPV in Nigeria did not seek help. Among those that sought help, only 1.9% sought help from formal sources compared to the 31.3% that sought help from informal sources. Moreover, Mengo, Sharma, and Beaujolais [12] reported that the majority (55.5%) of the women who experienced IPV in Kenya refused to seek help.

Evidence has shown that women are usually reluctant to disclose an experience of IPV in the beginning, mainly due to fear for themselves, fear for children, feelings of shame, denial, or fear of being judged by others [13]. Some women refuse to seek help for IPV because they perceive the violence as non-severe while others fail to either recognise the IPV or regard it as normal, mostly due to cultural norms [14]. The consequences of seeking help, such as worsened revictimization and fear that the abuser may harm family members or relatives may impede help-seeking among survivors of IPV [12]. Patriarchal socially shaped norms and values such as gender roles and expectations have also been reported to prevent women from seeking help for IPV. This is mainly because IPV is seen as a social norm and accepted within some African communities [15].

Women who are survivors of IPV usually seek help from their family and friends, which is help-seeking from the informal structures [14]. A study by Goodman et al. [8] reported that help-seeking from the informal setting presents a lower risk of re-abuse among women experiencing lower levels of IPV; however, it has no effect on those who experience higher levels of IPV. Women tend to seek help from people in the formal sectors including health professionals, crisis lines, ministers or clergy, shelters, and the criminal justice system (police, lawyers, etc.) when they persistently experience violence [16].

Studies have reported that violence characteristics and perception are strong predictors of help-seeking among women. For instance, the higher the severity of the IPV, the more help-seeking increases [14]. The form of violence experienced also predicts help-seeking. Women who are physically abused have an increased likelihood of seeking help compared to those who are abused emotionally or sexually [17]. Tenkorang, Owusu, and Kundhi [18] found in their study that women with a high-perceived risk of injury from IPV were more likely to seek help compared to those who saw themselves at no risk. Furthermore, the study found that respondents with high levels of trust in support services were likely to seek help as compared to those who did not trust those services. In Ethiopia, Muluneh, Alemu, and Meazaw [19] found that educational attainment, high wealth quintile, partner's employment status, partner's alcohol consumption status, and experience of physical violence significantly predicted women's help-seeking for IPV.

The high prevalence of IPV experienced by women in SSA is problematic, given that it is a violation of women's rights [2, 20]. To tackle the problem, several countries in SSA have implemented interventions to address the issue. In addition to the implementation of policies and laws to control IPV, there are increasing number of community-based interventions that aim to shift public opinion to the community level, mainly by addressing the traditional gender norms, roles, and expectations that are related to IPV [21]. These interventions have shown varying degrees of effectiveness in reducing the prevalence and preventing IPV [21]. A key intervention for survivors of IPV in several sub-Saharan African countries is the availability of support systems to reduce the health and social impacts of IPV [14, 22]. These support systems could be formal (such as speaking to a counsellor, seeking medical care, among others) or informal (such as speaking to a religious leader, family member, among others) [23]. The formal support usually begins by reporting the experience to the local domestic violence unit (example is the Domestic Violence and Victim Support Unit in Ghana) for them to link the survivor to the appropriate medical, psychological, and social support [24]. Even though the formal support systems have been shown to be effective in reducing and preventing negative health and social impact of IPV, the majority of women who survive IPV do not seek help from these support systems [10, 11]. The help-seeking behaviour of women who are exposed to IPV is understudied at the sub-Saharan African level. Few studies have examined the

predictors of IPV help-seeking among women in different sub-Saharan African countries. However, these studies used different datasets and statistical methods making it difficult to generalize the findings. This study, therefore, aimed at assessing the predictors of help-seeking among female survivors of IPV in SSA. Findings of this study could contribute to strengthening existing interventions that address IPV and enhance help seeking among women survivors of IPV in SSA. The study will also contribute to the realization of Sustainable Development Goals 3 (good health and well-being) and 5 (gender equality, including empowering women) [25].

## Materials and methods

### Data source and study design

We pooled data from the most recent Demographic and Health Surveys (DHS) of eighteen (18) countries in SSA. The data were extracted from the women's files in countries with datasets from 2014 to 2021. According to Corsi et al. [26], the DHS is a nationally representative survey conducted globally in over 85 low-and-middle-income countries. A cross-sectional design was adopted for the survey and a two-stage cluster sampling method was used to recruit the respondents for the survey. The first stage of sampling involved compiling a list of primary sampling units (PSUs) or enumeration areas (EAs) that covered the entire country and were obtained from the most recent national census. The EAs were further subdivided into standardized segments of 100–500 households each. Following that, a random sample of predetermined segment is chosen with a probability proportional to the size of the EA. At the second stage, households were systematically selected from a list of previously enumerated households in each selected EA segment, and those who were usual residents of selected households or visitors who slept in the households the night before the survey were interviewed. DHS used a structured questionnaire to collect data from the respondents on health and social indicators such as domestic violence [26]. In this study, 33,837 women in sexual relationships who had ever experienced IPV within the five years preceding the survey were included in the analysis (See Table 1). We prepared the manuscript following the guidelines outlined in the Strengthening Reporting of Observational Studies in Epidemiology (STROBE) [27]. The dataset is freely available for download at https://dhsprogram.com/data/available-datasets.cfm.

### Variables

**Outcome variable.** Help-seeking behaviour for IPV was the outcome variable. With this variable, only women who had ever experienced IPV were asked whether they sought help from someone. The response options were "sought help from someone" and "no help was sought". We recoded the responses into "Yes = 1" for those that responded 'sought help from someone and "No = 0" for those who said 'no help was sought'. The categorization used was based on literature [3, 28].

**Explanatory variables.** Nineteen explanatory variables were included in the study. These variables comprised individual-level and contextual-level factors. The individual-level variables included the woman's age (years), level of education, marital status, current working status, parity, person who usually decides on respondent's health care, person who usually decides on large household purchases, person who usually decides on visit to family or relatives, beating justified if wife goes out without telling husband, beating justified if wife neglects the children, Beating justified if wife argues with husband, beating justified if wife refuse to have sex with husband, beating justified if wife burns the food, exposure to interparental violence, frequency of watching television, frequency of listening to radio, and frequency of reading newspaper or magazine.

**Table 1. Description of study sample.**

| Countries | Year of survey | Weighted N | Weighted % |
|---|---|---|---|
| 1. Angola | 2015–16 | 3,015 | 8.91 |
| 2. Benin | 2017–18 | 1,271 | 3.76 |
| 3. Burundi | 2016–17 | 3,082 | 9.11 |
| 4. Cameroon | 2018 | 1,726 | 5.10 |
| 5. Ethiopia | 2016 | 1,263 | 3.73 |
| 6. Gambia | 2019–20 | 880 | 2.60 |
| 7. Liberia | 2019–20 | 1,058 | 3.13 |
| 8. Mali | 2018 | 1,564 | 4.62 |
| 9. Malawi | 2015–16 | 1,940 | 5.73 |
| 10. Nigeria | 2018 | 2,579 | 7.62 |
| 11. Rwanda | 2014–15 | 706 | 2.09 |
| 12. Sierra Leone | 2019 | 2,371 | 7.01 |
| 13. Chad | 2014–15 | 971 | 2.87 |
| 14. Tanzania | 2015–16 | 2,895 | 8.56 |
| 15. Uganda | 2016 | 3,521 | 10.41 |
| 16. South Africa | 2016 | 413 | 1.22 |
| 17. Zambia | 2018 | 2,545 | 7.52 |
| 18. Zimbabwe | 2015 | 2,037 | 6.02 |
| **All countries** | **2014–2020** | **33,837** | **100.00** |

We maintained the existing coding in the dataset for maternal age, current working status, and exposure to interparental violence. Marital status was recoded into 'married' and 'cohabiting'. The level of education was recoded into 'no education' 'primary', and 'secondary or higher'. Parity was recoded into 'zero birth', 'one birth', 'two births', 'three births', and 'four or more births'. For person who usually decides on respondent's health care, person who usually decides on large household purchases, and person who usually decides on visit to family or relatives, we maintained the existing responses, except the last two (someone else and other) which were merged to create one category. Hence, the response categories used in the study for the three variables were 'respondent alone', 'respondent and partner', 'partner alone', and 'someone else'. Similarly, we dropped those who responded "don't know" in these five variables on intimate partner violence justification (beating justified if wife goes out without telling husband, beating justified if wife neglects the children, beating justified if wife argues with husband, beating justified if wife refuse to have sex with husband, and beating justified if wife burns the food). Only those who responded 'yes' and 'no' were used in the study. In terms of the mass media variables (frequency of watching television, frequency of listening to radio, and frequency of reading newspaper or magazine), each of them was coded into 'not at all', 'less than once a week', and 'at least once a week'. Wealth index (poorest, poorer, middle, richer, richest), place of residence (urban, rural), and geographical sub-regions (Southern Africa, Central Africa, Eastern Africa, Western Africa) were the contextual-level variables in the study. All the explanatory variables used in the study were selected based on the significant association with the outcome variable from a theoretical perspective after reviewing of literature [3, 28–30].

## Statistical analyses

We carried out the data extraction, cleaning, and analysis using Stata version 16.0 (Stata Corporation, College Station, TX, USA). In all the analyses, we used the women's domestic violence model sampling weight (d005/1,000,000), clustering, and stratification as required by

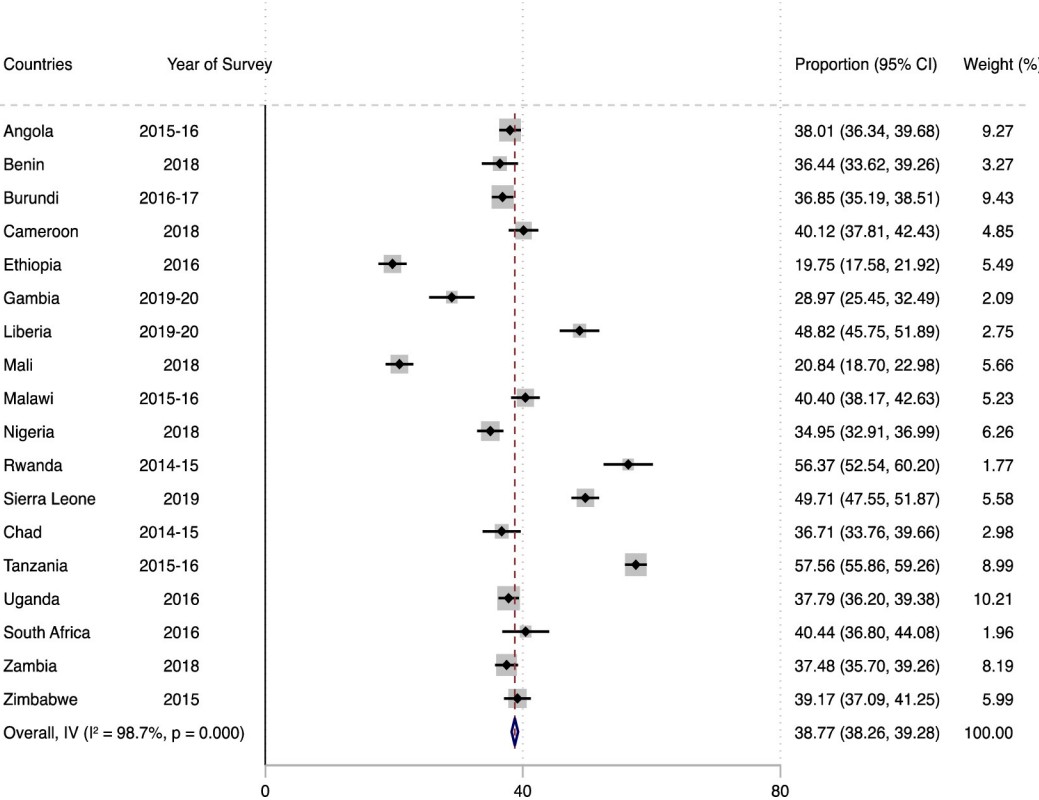

**Fig 1. Forest plot showing the prevalence of help seeking among survivors of intimate partner violence in sub-Saharan Africa.**

DHS to account for the complex survey design. The survey command "svyset" in Stata was used to declare the survey design, while all estimates were performed using the survey-specific command "svy". The analyses were conducted in three phases. In the first phase, a forest plot was used to present the prevalence of help-seeking behaviour among the survivors of IPV (Fig 1). In the second phase, Pearson chi-square test of independence was carried out to examine the distribution of help-seeking across the explanatory variables as well as determine the variables significantly associated with it (Table 2). To obtain the variables for the regression model, the best variable selection method was adopted. This method is used to select the combined set of variables that best explain the outcome variable under study [31, 32]. We estimated the best variables using the Stata command 'gvselect' and the output with the least Akaike's Information Criterion (AIC) with its corresponding set of variables was selected as the variables used in the regression model [31, 32]. Finally, using four models, we employed a multi-level binary regression analysis to examine the association between help-seeking and the explanatory variables (Model O—III). We fitted the first model (Model O) to examine the variance in help-seeking attributed to the clustering at the PSUs. Only individual-level variables were included in Model I. Model II included only contextual-level variables. The final model (Model III) included help-seeking and all the explanatory variables. The findings were presented in the form of adjusted odds ratios (AOR), along with their respective 95% confidence intervals (CIs). The AIC was used to assess model fitness and comparison, with the model with the lowest AIC being the best model. In the chi-square and regression analyses, statistical significance was set at $p < 0.05$.

**Table 2. Bivariable analysis of help-seeking behaviour among survivors of intimate partner violence in sub-Saharan Africa.**

| Variable | Weighted N | Weighted % | Sought help | | |
|---|---|---|---|---|---|
| | | | No (%) | Yes (%) | P-value |
| **Women's age (years)** | | | | | 0.018 |
| 15–19 | 1,845 | 5.4 | 69.0 | 31.0 | |
| 20–24 | 5,881 | 17.4 | 64.4 | 35.6 | |
| 25–29 | 7,341 | 21.7 | 64.1 | 35.9 | |
| 30–34 | 6,498 | 19.2 | 63.6 | 36.4 | |
| 35–39 | 5,539 | 16.4 | 62.7 | 37.3 | |
| 40–44 | 3,864 | 11.4 | 62.9 | 37.1 | |
| 45–49 | 2,868 | 8.5 | 62.8 | 37.2 | |
| **Educational level** | | | | | <0.001 |
| No education | 10,830 | 32.0 | 66.4 | 33.6 | |
| Primary | 13,350 | 39.5 | 60.9 | 39.1 | |
| Secondary or higher | 9,657 | 28.5 | 65.0 | 35.0 | |
| **Marital status** | | | | | <0.001 |
| Married | 25,155 | 74.3 | 64.9 | 35.1 | |
| Cohabiting | 8,682 | 25.7 | 60.7 | 39.3 | |
| **Current working status** | | | | | <0.001 |
| Not working | 9,499 | 28.1 | 68.2 | 31.8 | |
| Working | 24,338 | 71.9 | 62.1 | 37.9 | |
| **Parity** | | | | | <0.001 |
| Zero birth | 1,672 | 4.9 | 69.4 | 30.6 | |
| One birth | 4,447 | 13.2 | 65.8 | 34.2 | |
| Two birth | 5,623 | 16.6 | 64.2 | 35.8 | |
| Three births | 5,276 | 15.6 | 63.7 | 36.3 | |
| Four or more births | 16,819 | 49.7 | 62.7 | 37.3 | |
| **Person who usually decides on respondent's health care** | | | | | <0.001 |
| Respondent alone | 7,684 | 22.7 | 59.2 | 40.8 | |
| Respondent and partner | 14,126 | 41.8 | 64.8 | 35.2 | |
| Partner alone | 11,814 | 34.9 | 65.7 | 34.3 | |
| Someone else or other | 213 | 0.6 | 62.2 | 37.8 | |
| **Person who usually decides on large household purchases** | | | | | <0.001 |
| Respondent alone | 5,730 | 16.9 | 59.2 | 40.8 | |
| Respondent and partner | 14,481 | 42.8 | 65.2 | 34.8 | |
| Partner alone | 13,343 | 39.4 | 64.4 | 35.6 | |
| Someone else or other | 283 | 0.9 | 61.2 | 38.8 | |
| **Person who usually decides on visit to family or relatives** | | | | | <0.001 |
| Respondent alone | 7,563 | 22.4 | 59.8 | 40.2 | |
| Respondent and partner | 15,931 | 47.1 | 65.5 | 34.5 | |
| Partner alone | 10,165 | 30.0 | 64.2 | 35.8 | |
| Someone else or other | 178 | 0.5 | 59.9 | 40.1 | |
| **Beating justified if wife goes out without telling husband** | | | | | 0.962 |
| No | 21,967 | 64.9 | 63.8 | 36.2 | |
| Yes | 11,870 | 35.1 | 63.8 | 36.2 | |
| **Beating justified if wife neglects the children** | | | | | 0.579 |
| No | 20,580 | 60.8 | 64.0 | 36.0 | |
| Yes | 13,257 | 39.2 | 63.6 | 36.4 | |
| **Beating justified if wife argues with husband** | | | | | 0.585 |
| No | 22,205 | 65.6 | 63.7 | 36.3 | |

*(Continued)*

**Table 2.** (Continued)

| Variable | Weighted N | Weighted % | Sought help | | |
|---|---|---|---|---|---|
| | | | No (%) | Yes (%) | P-value |
| Yes | 11,632 | 34.4 | 64.1 | 35.9 | |
| **Beating justified if wife refuse to have sex with husband** | | | | | <0.001 |
| No | 23,724 | 70.1 | 62.8 | 37.2 | |
| Yes | 10,113 | 29.9 | 66.1 | 33.9 | |
| **Beating justified if wife burns the food** | | | | | <0.001 |
| No | 27,487 | 81.23 | 63.1 | 36.9 | |
| Yes | 6,350 | 18.77 | 67.1 | 32.9 | |
| **Exposed to interparental violence** | | | | | <0.001 |
| No | 21,785 | 64.4 | 65.9 | 34.1 | |
| Yes | 12,052 | 35.6 | 60.1 | 39.9 | |
| **Frequency of watching television** | | | | | 0.014 |
| Not at all | 20,945 | 61.9 | 63.5 | 36.5 | |
| Less than once a week | 4,291 | 12.7 | 62.0 | 38.0 | |
| At least once a week | 8,601 | 25.4 | 65.6 | 34.4 | |
| **Frequency of listening to radio** | | | | | 0.008 |
| Not at all | 14,455 | 42.7 | 64.7 | 35.3 | |
| Less than once a week | 6,550 | 19.4 | 61.7 | 38.3 | |
| At least once a week | 12,832 | 37.9 | 63.9 | 36.1 | |
| **Frequency of reading newspaper or magazine** | | | | | 0.015 |
| Not at all | 27,971 | 82.7 | 64.2 | 35.8 | |
| Less than once a week | 3,596 | 10.6 | 60.6 | 39.4 | |
| At least once a week | 2,270 | 6.7 | 64.3 | 35.7 | |
| **Wealth index** | | | | | 0.007 |
| Poorest | 6,701 | 19.8 | 62.4 | 37.6 | |
| Poorer | 7,134 | 21.1 | 62.8 | 37.2 | |
| Middle | 6,981 | 20.6 | 64.0 | 36.0 | |
| Richer | 6,811 | 20.1 | 63.6 | 36.4 | |
| Richest | 6,210 | 18.4 | 66.6 | 33.4 | |
| **Place of residence** | | | | | 0.799 |
| Urban | 11,449 | 33.8 | 64.0 | 36.0 | |
| Rural | 22,388 | 66.2 | 63.7 | 36.3 | |

## Ethical consideration

Ethical approval was not sought for this study since the DHS dataset is freely available in the public domain. Prior to the study, permission to use the dataset for publication was obtained from the MEASURE DHS. The detailed ethical guidelines is available at http://goo.gl/ny8T6X.

# Results

## Prevalence of help-seeking behaviour among female survivors in sub-Saharan Africa

Out of the 33,837 women who had experienced IPV in SSA, only 38.77% (95% CI = 38.26–39.28) of them sought help after experiencing IPV (see Fig 1). Ethiopia had the lowest prevalence of women who sought help after experiencing IPV (19.75%; 95% CI = 17.58–21.92) and Tanzania had the highest prevalence (57.56%; 95% CI = 55.86–59.26).

## Distribution of help-seeking behaviour across the explanatory variables

Table 2 shows the bivariable analysis of the explanatory variables considered in this study and their association with help seeking behaviour among survivors of IPV. In terms of age, help seeking behaviour peaked among women aged 35–39 (37.3%). However, it was lowest among those aged 15–19 (31.0%). With educational level, the highest proportion of help seeking (39.1%) was recorded among women with primary education whereas those with no formal education recorded the lowest (33.6%). Help seeking behaviour for IPV was prevalent among women who were cohabiting (39.3%). Regarding current working status, the highest proportion (37.9%) was recorded among those currently working. It was evident that women with parity four or more (37.3%) had the highest proportion of help seeking behaviour for IPV whereas those with parity zero had the lowest proportion (30.6%). Help seeking behaviour for IPV was also prevalent among women who usually decide on their health care (40.8%), those who usually decide on large household purchases alone (40.8%), and those who usually decide on visit to family or relatives alone (40.2%). High proportion of help seeking behaviour was also recorded among women who did not justify beating if wife refuses to have sex with husband (37.2%), those who did not justify beating if wife burns food (36.9%), and women who were exposed to inter-parental violence (39.9%). Moreover, women who watched television less than once a week (38.0%), those who listened to radio less than once a week (38.3%), those who read newspaper or magazine less than once a week (39.4%), and those who were poorest (37.6%) had the highest prevalence of help seeking behaviour for IPV. Finally, the chi square analysis shows that all explanatory variables were significantly associated with help seeking behaviour except justification of beating if wife goes out without telling husband, if wife neglects the children, if wife argues with husband, and place of residence (Table 2).

## Predictors of help seeking behaviour among female survivors of IPV in sub-Saharan Africa

Model III of Table 3 shows the multilevel analysis of factors associated with help seeking behaviour for IPV among women in SSA. It was found that the odds of seeking help for IPV was higher among women who were cohabiting [AOR = 1.18, 95% CI = 1.09, 1.27] compared to those married. Women with primary education [AOR = 1.18, 95% CI = 1.09, 1.27] had higher odds of seeking help for IPV compared to those with no formal education. In terms of current working status, women currently working were more likely to seek help for IPV [AOR = 1.27, 95% CI = 1.17, 1.38] relative to those who were not working. The odds of seeking help for IPV increased with increasing level of parity. Specifically, women with four or more children had the highest odds of seeking help for IPV [AOR = 1.32, 95% CI = 1.13, 1.54] compared to those without a child. Women who were exposed to inter-parental violence had a higher likelihood of seeking help for IPV [AOR = 1.25, 95% CI = 1.18, 1.34] relative to those who were not exposed to inter-parental violence. Women whose primary health care decision was jointly taken with the partner [AOR = 0.81, 95 CI = 0.75, 0.88] and those whose partners usually decide on their health care alone [AOR = 0.83, 95% CI = 0.76, 0.91] had lower odds of seeking help for IPV compared to those who usually decide on their health care alone. Women who listened to radio [AOR = 1.10, 95% CI = 1.01, 1.20], and those who read newspaper or magazine [AOR = 1.14, 95% CI = 1.02, 1.28] less than once a week were more likely to seek help for IPV than those who did not. For IPV justification, women who justified beating if wife argues with husband [AOR = 1.11, 95% CI = 1.02, 1.21], those who justified beating if wife refuses to have sex with husband [AOR = 0.84, 95 CI = 0.77, 0.92], and women who justified beating if wife burns the food [AOR = 0.83, 95 CI = 0.76, 0.90] were less likely to seek help for IPV compared to those who did not justify beating. Also, the odds of seeking help for IPV was

**Table 3. Fixed and random effect analysis of predictors of help-seeking behaviour for IPV among women in sub-Saharan Africa.**

| Variables | Model O | Model I aOR [95% CI] | Model II aOR [95% CI] | Model III aOR [95% CI] |
|---|---|---|---|---|
| **Fixed effect results** | | | | |
| **Marital status** | | | | |
| Married | | 1.00 | | 1.00 |
| Cohabiting | | 1.18*** [1.09,1.27] | | 1.18*** [1.09,1.27] |
| **Educational level** | | | | |
| No education | | 1.00 | | 1.00 |
| Primary | | 1.19*** [1.11,1.29] | | 1.18*** [1.09,1.27] |
| Secondary or higher | | 1.06 [0.96,1.16] | | 1.07 [0.96,1.18] |
| **Current working status** | | | | |
| Not working | | 1.00 | | 1.00 |
| Working | | 1.26*** [1.16,1.36] | | 1.27*** [1.17,1.38] |
| **Parity** | | | | |
| Zero birth | | 1.00 | | 1.00 |
| One birth | | 1.17 [0.98,1.40] | | 1.17 [0.98,1.39] |
| Two births | | 1.23* [1.04,1.45] | | 1.22* [1.03,1.44] |
| Three births | | 1.279** [1.09,1.50] | | 1.28** [1.09,1.50] |
| Four or more births | | 1.32*** [1.13,1.54] | | 1.32*** [1.13,1.54] |
| **Exposed to interparental violence** | | | | |
| No | | 1.00 | | 1.00 |
| Yes | | 1.26*** [1.19,1.34] | | 1.25*** [1.18,1.34] |
| **Person who usually decides on respondent's health care** | | | | |
| Respondent alone | | 1.00 | | 1.00 |
| Respondent and partner | | 0.80*** [0.74,0.87] | | 0.81*** [0.75,0.88] |
| Partner alone | | 0.81*** [0.74,0.88] | | 0.83*** [0.76,0.91] |
| Someone else or other | | 1.01 [0.68,1.51] | | 1.02 [0.69,1.53] |
| **Frequency of watching television** | | | | |
| Not at all | | 1.00 | | 1.00 |
| Less than once a week | | 1.03 [0.93,1.14] | | 1.05 [0.94,1.17] |
| At least once a week | | 0.89* [0.82,0.98] | | 0.94 [0.85,1.04] |
| **Frequency of listening to radio** | | | | |
| Not at all | | 1.00 | | 1.00 |
| Less than once a week | | 1.09* [1.00,1.19] | | 1.10* [1.01,1.20] |
| At least once a week | | 0.99 [0.91,1.07] | | 0.99 [0.92,1.08] |
| **Frequency of reading newspaper or magazine** | | | | |
| Not at all | | 1.00 | | 1.00 |
| Less than once a week | | 1.14* [1.02,1.28] | | 1.14* [1.02,1.28] |
| At least once a week | | 1.04 [0.90,1.20] | | 1.04 [0.90,1.21] |
| **Beating justified if wife goes out without telling husband** | | | | |
| No | | 1.00 | | 1.00 |
| Yes | | 1.08 [1.00,1.18] | | 1.08 [1.00,1.18] |
| **Beating justified if wife argues with husband** | | | | |
| No | | 1.00 | | 1.00 |
| Yes | | 1.12* [1.02,1.22] | | 1.11* [1.02,1.21] |
| **Beating justified if wife refuse to have sex with husband** | | | | |
| No | | 1.00 | | 1.00 |
| Yes | | 0.85*** [0.77,0.93] | | 0.84*** [0.77,0.92] |
| **Beating justified if wife burns the food** | | | | |

*(Continued)*

**Table 3.** (Continued)

| Variables | Model O | Model I aOR [95% CI] | Model II aOR [95% CI] | Model III aOR [95% CI] |
|---|---|---|---|---|
| No | | 1.00 | | 1.00 |
| Yes | | 0.83*** [0.76,0.90] | | 0.83*** [0.76,0.90] |
| **Wealth index** | | | | |
| Poorest | | | 1.00 | 1.00 |
| Poorer | | | 0.99 [0.91,1.08] | 0.97 [0.89,1.06] |
| Middle | | | 0.92 [0.84,1.02] | 0.91 [0.83,1.01] |
| Richer | | | 0.92 [0.83,1.02] | 0.92 [0.83,1.03] |
| Richest | | | 0.74*** [0.65,0.84] | 0.77*** [0.67,0.89] |
| **Place of residence** | | | | |
| Urban | | | 1.00 | 1.00 |
| Rural | | | 0.87** [0.79,0.96] | 0.89* [0.81,0.99] |
| **Geographical sub-regions** | | | | |
| Southern Africa | | | 1.00 | 1.00 |
| Central Africa | | | 0.92 [0.81,1.04] | 0.90 [0.78,1.03] |
| Eastern Africa | | | 1.08 [0.98,1.19] | 0.98 [0.88,1.09] |
| Western Africa | | | 0.87** [0.79,0.96] | 0.92 [0.82,1.04] |
| **Random effects** | | | | |
| PSU variance (95% CI) | 0.09 [0.07, 0.122] | 0.09 [0.07, 0.12] | 0.10 [0.08, 0.13] | 0.09 [0.07, 0.12] |
| ICC | 0.028 | 0.027 | 0.029 | 0.028 |
| Wald chi-square | Reference | 293.99 (<0.001) | 47.03 (<0.001) | 317.77 (<0.001) |
| **Model fitness** | | | | |
| Log-likelihood | -20489.92 | -20247.393 | -20444.586 | -20229.009 |
| AIC | 40983.84 | 40542.79 | 40909.17 | 40522.02 |
| N | 33,837 | 33,837 | 33,837 | 33,837 |
| Number of clusters | 1,289 | 1,289 | 1,289 | 1,289 |

aOR = adjusted odds ratios; CI = Confidence Interval

* $p < 0.05$

** $p < 0.01$

*** $p < 0.001$; 1 = Reference category; PSU = Primary Sampling Unit; ICC = Intra-Class Correlation; AIC = Akaike's Information Criterion; N = Total sample

low among women in the richest wealth quintile [AOR = 0.77, 95% CI = 0.67, 0.89] relative to those in the poorest wealth quintile. Finally, women living in rural communities were less likely to seek help for IPV [AOR = 0.89, 95% CI = 0.81, 0.99] compared to those in urban communities.

## Discussion

This study examined the prevalence and predictors of help-seeking behaviour for IPV among women in 18 countries in SSA. The study found that the overall prevalence of help seeking for IPV was 38.77% with Ethiopia and Tanzania recording the lowest (19.75%) and highest (57.56%) prevalence respectively. This shows that the majority of women in SSA (approximately 61%) did not seek help after experiencing IPV. This finding corroborates findings of previous studies conducted in low-and middle-income countries including Nigeria [18, 33, 34], India [3, 35], China [36], and the Gambia [28]. Also, our findings contrast those of previous research conducted in high-income countries, which showed that the majority of women asked for assistance or help after experiencing IPV [16, 37]. The percentage of women in Tanzania who seek

help for IPV could be due to the presence of both formal (police, hospitals, legal services, and social protection-shelters) and informal (family members, religious leaders) support systems. The high rate of help-seeking among these women is a result of the country's support systems being readily available, as seen by the high prevalence of such behavior among them [38].

The low percentage of help-seeking observed among Ethiopian and Malian women could be due to several factors encompassing economic, social, and cultural. For instance, the cash transfer programme, which was implemented in 2014 that only targeted male household heads, could have contributed to the low help-seeking rate by empowering only men at the detriment of women [39]. Again, the low percentage of women seeking help in SSA could be explained by the fact that some African societies regard IPV as a family and private issue, because of family and community norms. Women in these societies are expected to be virtuous and submissive spouses who sacrifice themselves for their families [19, 40, 41]. This puts women in an awkward situation if their husband is abusive. Additionally, most women believe that they can handle IPV situations on their own [36]. For some survivors, going to formal channels to report their experiences goes against generally known social standards of "not washing one's dirty linen in public" [42]. Moreover, the low help-seeking among survivors of IPV in Mali and Ethiopia could be attributed to the societal assertion that IPV is a family matter and should be dealt with secretly is a barrier to alleviating IPV from such societies. Often, women are supposed to be well-behaved and obedient wives and mothers, to give of themselves for their families, and to keep family affairs private. Should their husband become aggressive, this puts them in a challenging situation. Also, some women might decide to remain silent about their ordeal in order to preserve the appearance of a happy household. Additionally, the issues of African kingship system, cultural norms, and religious background may further contribute to the low degree of IPV help seeking because family concerns are not typically discussed outside of marriage or romantic relationships [2].

Marital status was found to be a substantial factor of help-seeking behaviour for IPV among women in SSA. Women who were cohabiting had higher odds of help seeking compared to those who were married. Linos et al. [34] and Parvin et al. [43] found that women who were not in officially married were more likely to seek help for IPV compared to those who were married. One probable explanation is because unmarried women are not bound by marriage legal concerns and can quickly leave a relationship. As a result, they would not be in a contemplative state but would rather act quickly by seeking external IPV help [37]. This finding, however, contradicts Meyer's [37] conclusion that married women who had been subjected to IPV were more likely to seek assistance to cease the abuse. Thus, survivors of IPV are more likely to actively seek help to cease the violence when they are in a marital relationship.

This study showed that women's educational level is a predictor of help-seeking behaviour for IPV. Women with a primary level of education showed a greater probability of seeking help relative to those without education. Our finding supports other findings [3, 43, 44] which indicated that an increase in the educational level increases help seeking behaviour for IPV. This could be attributed to the fact that education empowers and enlightens women to easily access information, especially legal information that could facilitate their IPV help seeking. Previous studies argue that formal education enlighten women to recognize that IPV is inappropriate, understand their rights, and know where to seek assistance [45, 46]. However, our findings are inconsistent with that of Rowan et al. [47], who found that women with no education had a higher likelihood of help seeking for IPV than those with higher levels of education in India. This could be owing to India's strict societal conventions, which more educated women must adhere to, prompting them to remain mute regarding IPV. Other factors that hinder educated women from getting help include psychological or cognitive hurdles such as fear and stigma [48].

A woman's working status was strongly associated with seeking help for IPV. Specifically, women who were currently working were more likely to seek help for IPV. This finding supports other findings [28, 34, 49–51] where a woman's help seeking behaviour was connected to her working status. This finding could be explained in several pathways. One probable reason is that working women are able to earn wages that allow them to accumulate greater social and financial resources which make them more proactive in identifying proper sources of aid to seek help for IPV [34, 44]. Besides, working women are thought to be financially self-sufficient which could make them more likely to request assistance. However, our findings contradict those of Hu et al. [36] and Rocca et al. [52], who found that working women had lower likelihood of seeking care for IPV.

In SSA, women's parity was associated with help seeking for IPV. Our findings revealed that women with two or more children were more likely to seek help for IPV relative to women with no children. The significant link between parity and assistance-seeking behavior is comparable to the findings of Parvin et al. [43] and Meyer [37], who found that the existence of children increased the probability of help seeking behaviour among IPV survivors. One possible reason is that women seek help because they are worried about their children's safety. While survivors may not always pursue the support they need to defend themselves, the possibility of their children witnessing the violence enhances their chances of getting help [37]. This explanation is backed by a WHO multi-country research revealing that in Bangladesh, about one-third of women who suffered physical violence sought help because their children were warned or beaten by the abuser [43].

Exposure to interparental violence was associated with seeking help for IPV. Previous studies [28, 34, 47, 53] found that women who watched their fathers abusing or beating their mothers were more likely to seek help to prevent the violence. According to Rowan and colleagues [47], having health care experts inquire about abuse experienced as a child could be a good place to start a discussion regarding IPV. This is because women are emotionally bonded to their mothers; they view their fathers' violence as unacceptable and, as a result, seek support from outside sources due to the inability of the woman's family to help [54]. It is also possible that women seek support because of concerns about their mothers' health. However, other studies have found that women who have been exposed to IPV justify it and are, therefore, less probable to seek help for it [55, 56].

This study found that women who listened to radio and read newspaper or magazine less than once a week were more likely to seek help compared to those who did not. Exposure to both electronic and print media helps women to get information about their rights as wives and how to fight for these rights. Mass media also educate and inform women about what constitutes IPV and how and where to seek help when they find themselves in IPV situations [57]. Our results also found persons who usually decide on respondents' healthcare to influence help seeking behaviour for IPV. Thus, lower odds of help seeking behaviour was recorded among women who took a decision concerning their healthcare together with the partner, and those who decided on partner's health care alone compared to women who usually decide on their healthcare alone [47].

Our study found that justifying IPV reduces the likelihood of seeking help among women in SSA, which is consistent with our findings. Women who justified beating if wife argues with husband, if wife refuses to have sex with husband and if wife burns the food had a lower likelihood of seeking help than those who did not. Previous studies have shown that women who believe men are right in assaulting or hitting their wives in any situation are less likely to ask for help because of their notion that a husband has the liberty to be aggressive against his wife [3, 58, 59]. The fact that women rationalize IPV shows that conventional gender norms governing women's roles in the home are likely to guide sub-Saharan African women and their

actions. IPV may have been perceived as a routine response for these women when they fail to fulfill their culturally defined gender norms. On the other hand, women who do not accept IPV are less likely to regard violence as acceptable behaviour and may be less influenced by the extreme gender beliefs that influence marital relations in the region [60].

In our analysis, wealth status was found to have an inverse relation with IPV help seeking behavior. Women in the richest wealth quintile were less likely to seek care for IPV compared to those in the poorest wealth quintile. This suggests that help seeking behaviour reduces when there is an increase in wealth status. However, the link between IPV and help-seeking behavior is inconclusive. While some studies have discovered a favorable link, others have found the opposite. The findings of this study are consistent with previous studies conducted in SSA [34] and Asia [61]. According to Tichy et al. [61], in India, women with the highest wealth status were less likely than those with the lowest wealth status to notice abuse. This could be that affluence is linked to violent underreporting. Our findings, however, contradict those of Muluneh et al. [19] and Kim and Gray [62] in an Ethiopian and American study respectively.

This study also showed a statistically significant association between place of residence and help seeking behaviour for IPV. Women in rural areas reported lower odds of help-seeking behaviour compared to women in urban settings. The reason might be that women in rural locations have less or no formal support services where they could seek help for IPV than women in urban areas. Women in rural communities are also expected to seek assistance only from their partners' families, as they are the ones who resolve marital issues [63]. Many people in rural areas believe that family disputes should be private or kept hidden. They feel that because women are socialized as homemakers and family gatekeepers, they have a unique responsibility to play in safeguarding these values [11]. While these socio-cultural norms are occasionally beneficial, they have formed blockades to women seeking assistance in rural areas [18].

## Strengths and limitations

This study drew on a considerable amount of data from nationally representative samples across various countries in SSA, which improved the accuracy and generalizability of the findings. However, the study's conclusions are constrained in some ways. First, the study used cross-sectional data, which limits causal interpretations of the results. Moreover, since the outcome variable is self-reported, recall bias may exist and be underreported due to fear of stigma and prejudice, which cannot be verified by formal or informal sources. Also, because the survey questions were so sensitive, not all women who had been subjected to spousal abuse likely reported it—a problem that always arises when conducting IPV research. Additionally, the data used in this study were restricted to women only, which is in line with the widely held assumption that women are the most common survivors of IPV. While this widely held assumption is not always the case, this study provides timely and vital information that can be used to increase help-seeking behaviour for IPV among women in SSA.

## Policy and practice implications

The results of this and other similar research show only a small percentage of women in SSA seek support for IPV. This necessitates taking steps to ensure that there are sufficient formal and informal sources of assistance. Family, friends, and neighbors can often provide immediate assistance in the form of food and shelter, and can intervene in dealing with the violence [64], but they are rarely able to assist the survivors in changing her circumstances in the long run [3]. Despite the fact that IPV happens at home, it is critical that violence is not treated as a personal affair. IPV should be viewed as a political and social issue with multiple levels of involvement required. There is the need to ensure that all children receive at least a primary,

but preferably a secondary level education. This is because education helps in empowering girls, which has the tendency to increase future help-seeking rates for IPV. Through media programs targeting parents, the long-term profits of females enrolling in school could be achieved. However, concentrating on individual measures to strengthen women's empowerment may not bring a significant rise in help-seeking as far as patriarchal attitudes towards violence continue to exist [54]. Therefore, it is critical to address both violence-related norms and attitudes that prevent women from reporting violence to security and legal authorities for necessary action. Based on the findings, public awareness would help to address the consequences of IPV. Respective government must increase their coverage of formal support services to IPV survivors especially those in rural communities. Additionally, countries reporting low prevalence of help-seeking should leverage on the MAISHA Intervention and the economic and social empowerment initiatives in Tanzania to aid deal with IPV [65, 66].

## Conclusion

This study has proffered significant insights into the predictors of help-seeking behavior for IPV among women in SSA. The overall prevalence of help-seeking behavior for IPV was found to be low. Women in Mali had the lowest prevalence of help-seeking behavior in SSA while women in Tanzania had the highest prevalence. Factors associated with help seeking behaviour for IPV include marital status, educational level, working status, parity, exposure to interparental violence, mass media (radio, newspaper or magazine), IPV justification, wealth index, and place of residence. Since seeking help is both desirable and crucial, we urge policymakers to address the hurdles that prevent women from getting help, including establishing IPV service programs for female survivors. It is also critical to educate both women and men about the dangers of rationalizing IPV. It is important that state institutions empower unemployed women to be financially and socially independent.

## Author Contributions

**Conceptualization:** Richard Gyan Aboagye, Abdul-Aziz Seidu, Abdul Cadri, Tarif Salihu, Francis Arthur-Holmes, Sarah Tara Sam, Bright Opoku Ahinkorah.

**Data curation:** Richard Gyan Aboagye, Abdul-Aziz Seidu, Bright Opoku Ahinkorah.

**Formal analysis:** Richard Gyan Aboagye, Abdul-Aziz Seidu, Bright Opoku Ahinkorah.

**Methodology:** Richard Gyan Aboagye, Abdul-Aziz Seidu, Abdul Cadri, Bright Opoku Ahinkorah.

**Software:** Richard Gyan Aboagye, Abdul-Aziz Seidu, Bright Opoku Ahinkorah.

**Validation:** Abdul-Aziz Seidu, Abdul Cadri, Tarif Salihu, Francis Arthur-Holmes, Sarah Tara Sam, Bright Opoku Ahinkorah.

**Visualization:** Richard Gyan Aboagye.

**Writing – original draft:** Richard Gyan Aboagye, Abdul-Aziz Seidu, Abdul Cadri, Tarif Salihu, Francis Arthur-Holmes, Sarah Tara Sam, Bright Opoku Ahinkorah.

**Writing – review & editing:** Richard Gyan Aboagye, Abdul-Aziz Seidu, Abdul Cadri, Tarif Salihu, Francis Arthur-Holmes, Sarah Tara Sam, Bright Opoku Ahinkorah.

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
