## [Decision Letter · Decision Letter 0]

21 Dec 2022

PONE-D-21-35858Ending violence against women and girls: help seeking behaviour of women exposed to intimate partner violence in sub-Saharan AfricaPLOS ONE

Dear Dr. Aboagye,

Thank you for submitting your manuscript to PLOS ONE. I sincerely apologise for the unusually delayed review timeframe. Your manuscript has been assessed by one reviewer, whose comments are appended below. After careful consideration, we feel that it has merit but does not fully meet PLOS ONE’s publication criteria as it currently stands. Therefore, we invite you to submit a revised version of the manuscript that addresses the points raised during the review process. In addition to the comments raised by the reviewer, please ensure that you update the introduction and discussion to reference any relevant literature that has been published since this study was submitted. Please note that we have only been able to secure a single reviewer to assess your manuscript. We are issuing a decision on your manuscript at this point to prevent further delays in the evaluation of your manuscript. Please be aware that the editor who handles your revised manuscript might find it necessary to invite additional reviewers to assess this work once the revised manuscript is submitted. However, we will aim to proceed on the basis of this single review if possible.

We look forward to receiving your revised manuscript.

Kind regards,

Emily Chenette

Editor in Chief

PLOS ONE

Journal Requirements:

Reviewers' comments:

Reviewer's Responses to Questions

**Comments to the Author**

1. Is the manuscript technically sound, and do the data support the conclusions?

Reviewer #1: Yes

2. Has the statistical analysis been performed appropriately and rigorously? 

Reviewer #1: Yes

3. Have the authors made all data underlying the findings in their manuscript fully available?

Reviewer #1: Yes

4. Is the manuscript presented in an intelligible fashion and written in standard English?

Reviewer #1: Yes

5. Review Comments to the Author

Reviewer #1: The paper is an interesting read and quite well written. However, there are few points that authors might want to consider:

1. Lines 137-138 mention about interventions that were introduced in Sub Saharan Africa to deal with the problem of intimate partner violence. Authors should elaborate on the nature and enforcements of these interventions.

2. Lines 139-140 highlights that the key intervention is the support system for women facing IPV. Authors need to explain whether this support system is formal or informal? Moreover, if there exists a formal support system, discuss its implementation and effectiveness.

3. Some information regarding the proportion of females experiencing IPV in each country must be presented and its connection with the proportion of women seeking out support should be discussed.

4. According to Figure 1, 35.89% of women experiencing IPV in SSA sought for help. Table 2 reveals that approximately one third of the women corresponding to almost all sub categories sought help after experiencing IPV. So pooling of cross sectional data across SSA reflects a monotonous sort of finding. The analysis can be made interesting by adding country specific details especially of those with higher/lower prevalence than average values. Pooling cross sections must present some details that are normally missed out in country specific analysis.

5. Mali and Tanzania experience lowest and highest prevalence of help seeking. Relate these findings to the specific support mechanisms available for females.

6. PLOS authors have the option to publish the peer review history of their article (what does this mean?). If published, this will include your full peer review and any attached files.

Reviewer #1: No

---

## [Author Response · Author response to Decision Letter 0]

5 Apr 2023

RESPONSE TO REVIEWERS COMMENTS

Reviewer #1: The paper is an interesting read and quite well written. However, there are few points that authors might want to consider:

1. Lines 137-138 mention about interventions that were introduced in Sub Saharan Africa to deal with the problem of intimate partner violence. Authors should elaborate on the nature and enforcements of these interventions.

Response: Thank you. We have addressed this comment.

2. Lines 139-140 highlights that the key intervention is the support system for women facing IPV. Authors need to explain whether this support system is formal or informal? Moreover, if there exists a formal support system, discuss its implementation and effectiveness.

Response: Thank you. The kind of support systems has been indicated, as well as its implementation and effectiveness. 

3. Some information regarding the proportion of females experiencing IPV in each country must be presented and its connection with the proportion of women seeking out support should be discussed.

Response: Thank you. We have addressed this comment.

4. According to Figure 1, 35.89% of women experiencing IPV in SSA sought for help. Table 2 reveals that approximately one third of the women corresponding to almost all sub categories sought help after experiencing IPV. So pooling of cross sectional data across SSA reflects a monotonous sort of finding. The analysis can be made interesting by adding country specific details especially of those with higher/lower prevalence than average values. Pooling cross sections must present some details that are normally missed out in country specific analysis.

Response: Thank you. We have provided supplementary containing results for the predictors of help seeking for IPV among women in Mali and Tanzania (Table S1).

5. Mali and Tanzania experience lowest and highest prevalence of help seeking. Relate these findings to the specific support mechanisms available for females.

Response: Thank you. We have addressed this comment.

---

## [Decision Letter · Decision Letter 1]

6 Sep 2023

PONE-D-21-35858R1Ending violence against women and girls: help seeking behaviour of women exposed to intimate partner violence in sub-Saharan AfricaPLOS ONE

Dear Dr. Aboagye,

Thank you for submitting your manuscript to PLOS ONE. After careful consideration, we feel that it has merit but does not fully meet PLOS ONE’s publication criteria as it currently stands. Therefore, we invite you to submit a revised version of the manuscript that addresses the points raised during the review process.

The authors have been responsive to previous comments. A few additional notes:

In the Abstract (Introduction), the authors wrote, “We assessed the prevalence and predictors of help-seeking among female victims of IPV in sub-Saharan Africa.” Given the cross-sectional nature of the data, I suggest the authors refer to “factors associated with help-seeking” rather than “predictors of help-seeking” throughout the manuscript.

Language – I suggest the authors do thorough copy-editing of the manuscript. For instance, in the Abstract (Introduction), the authors wrote, “The data was extracted …” rather than “The data **were **extracted ...”

Lines 461-472: Beyond acceptance of IPV and the referenced conditional cash transfers (for Mali), how culturally acceptable is help-seeking for IPV in Mali and Ethiopia? Are there other sociocultural factors in these settings that may help to further explain the poor help-seeking behavior documented in this study?

Line 508 should read, “Policy and Programmatic Implications” as both are addressed in the paragraph that follows. Are there particular policies in some of these settings that the authors would like to call out? Any exemplars, especially where help-seeking for IPV is high?

We look forward to receiving your revised manuscript.

Kind regards,

Funmilola M. OlaOlorun, PhD

Academic Editor

PLOS ONE

Journal Requirements:

Reviewers' comments:

Reviewer's Responses to Questions

**Comments to the Author**

1. If the authors have adequately addressed your comments raised in a previous round of review and you feel that this manuscript is now acceptable for publication, you may indicate that here to bypass the “Comments to the Author” section, enter your conflict of interest statement in the “Confidential to Editor” section, and submit your "Accept" recommendation.

Reviewer #2: All comments have been addressed

2. Is the manuscript technically sound, and do the data support the conclusions?

Reviewer #2: Yes

3. Has the statistical analysis been performed appropriately and rigorously? 

Reviewer #2: Yes

4. Have the authors made all data underlying the findings in their manuscript fully available?

Reviewer #2: Yes

5. Is the manuscript presented in an intelligible fashion and written in standard English?

Reviewer #2: Yes

6. Review Comments to the Author

Reviewer #2: As stated by a previous reviewer, this manuscript is well written and the authors have made a good attempt at addressing the comments of the previous reviewers().

However, the following comments are to improve the readability of the manuscript:

a. Page 7, line 238-240: The lines contain two statements that seem repetitive, the authors should consider rephrasing.

b. Page 16, line 376-379: The authors seem to suggest that the failure of the cash transfer programme to include women was the main reason for the low help seeking behaviour of women in Mali. The cash transfer programme may have made some contributions to this problem, but it should not be the main reason for the low help seeking behaviour of IPV victims. The authors should consider rephrasing the statement and highlighting other possible reasons, which may include the socio-economic peculiarities of Mali.

7. PLOS authors have the option to publish the peer review history of their article (what does this mean?). If published, this will include your full peer review and any attached files.

Reviewer #2: No

---

## [Author Response · Author response to Decision Letter 1]

7 Sep 2023

Response to Comments

The authors have been responsive to previous comments. A few additional notes: In the Abstract (Introduction), the authors wrote, “We assessed the prevalence and predictors of help-seeking among female victims of IPV in sub-Saharan Africa.” Given the cross-sectional nature of the data, I suggest the authors refer to “factors associated with help-seeking” rather than “predictors of help-seeking” throughout the manuscript.

Response: We have changed ‘predictors of help-seeking’ to ‘factors associated with help-seeking’ throughout the manuscript.

Language – I suggest the authors do thorough copy-editing of the manuscript. For instance, in the Abstract (Introduction), the authors wrote, “The data was extracted …” rather than “The data were extracted ...”

Response: We have changed ‘The data was extracted’ to ‘The data were extracted’.

Lines 461-472: Beyond acceptance of IPV and the referenced conditional cash transfers (for Mali), how culturally acceptable is help-seeking for IPV in Mali and Ethiopia? Are there other sociocultural factors in these settings that may help to further explain the poor help-seeking behavior documented in this study?

Response: Thank you. The societal assertion that IPV is a family matter and should be dealt with secretly is a barrier to alleviating IPV from such societies. Often, women are supposed to be well-behaved and obedient wives and mothers, to give of themselves for their families, and to keep family affairs private. Should their husband become aggressive, this puts them in a challenging situation. Also, some women might decide to remain silent about their ordeal in order to preserve the appearance of a happy household. Additionally, the issues of African kingship system, cultural norms, and religious background may further contribute to the low degree of IPV help seeking because family concerns are not typically discussed outside of marriage or romantic relationships [2].

Line 508 should read, “Policy and Programmatic Implications” as both are addressed in the paragraph that follows. Are there particular policies in some of these settings that the authors would like to call out? Any exemplars, especially where help-seeking for IPV is high?

Response: Thank you. We have listed some interventions / initiatives employed in Tanzania to deal with IPV such that other countries should adopt their methodologies and recommendations.

Reviewer #2: As stated by a previous reviewer, this manuscript is well written and the authors have made a good attempt at addressing the comments of the previous reviewers ().

However, the following comments are to improve the readability of the manuscript:

a. Page 7, line 238-240: The lines contain two statements that seem repetitive, the authors should consider rephrasing.

Response: Thank you. We have corrected this error.

b. Page 16, line 376-379: The authors seem to suggest that the failure of the cash transfer programme to include women was the main reason for the low help seeking behaviour of women in Mali. The cash transfer programme may have made some contributions to this problem, but it should not be the main reason for the low help seeking behaviour of IPV victims. The authors should consider rephrasing the statement and highlighting other possible reasons, which may include the socio-economic peculiarities of Mali.

Response: Thank you. We have provided additional reasons, which might have accounted for the low prevalence of help-seeking for IPV victims.

---

## [Editor Report · Decision Letter 2]

10 Sep 2023

Ending violence against women and girls: help seeking behaviour of women exposed to intimate partner violence in sub-Saharan Africa

PONE-D-21-35858R2

Dear Dr. Aboagye,

We’re pleased to inform you that your manuscript has been judged scientifically suitable for publication and will be formally accepted for publication once it meets all outstanding technical requirements.

Kind regards,

Funmilola M. OlaOlorun, PhD

Academic Editor

PLOS ONE
---

## [Editor Report · Acceptance letter]

25 Sep 2023

PONE-D-21-35858R2 

Ending violence against women: help-seeking behaviour of women exposed to intimate partner violence in sub-Saharan Africa 

Dear Dr. Aboagye:

I'm pleased to inform you that your manuscript has been deemed suitable for publication in PLOS ONE. Congratulations! Your manuscript is now with our production department. 

Kind regards, 

on behalf of

Dr. Funmilola M. OlaOlorun 

Academic Editor

PLOS ONE